# Antidiabetic Effect of Millet Bran Polysaccharides Partially Mediated via Changes in Gut Microbiome

**DOI:** 10.3390/foods11213406

**Published:** 2022-10-28

**Authors:** Jinhua Zhang, Wenjing Wang, Dingyi Guo, Baoqing Bai, Tao Bo, Sanhong Fan

**Affiliations:** 1College of Life Sciences, Shanxi University, Taiyuan 030006, China; 2Shanxi Key Laboratory of Research and Utilization of Characteristic Plant Resources, Shanxi University, Taiyuan 030006, China

**Keywords:** millet bran polysaccharide (MBP), 16S rRNA gene sequencing, type 2 diabetic

## Abstract

Diabetes is a type of metabolic disease associated with changes in the intestinal flora. In this study, the regulatory effect of millet bran on intestinal microbiota in a model of type 2 diabetes (T2DM) was investigated in an effort to develop new approaches to prevent and treat diabetes and its complications in patients. The effect of purified millet bran polysaccharide (MBP) with three different intragastric doses (400 mg/kg, 200 mg/kg, and 100 mg/kg) combined with a high-fat diet was determined in a streptozotocin (STZ)-induced model of T2DM. By analyzing the changes in indicators, weight, fasting blood sugar, and other bio-physiological parameters, the changes in gut microbiota were analyzed via high-throughput sequencing to establish the effect of MBP on the intestinal flora. The results showed that MBP alleviated symptoms of high-fat diet-induced T2DM. A high dosage of MBP enhanced the hypoglycemic effects compared with low and medium dosages. During gavage, the fasting blood glucose (FBG) levels of rats in the MBP group were significantly reduced (*p* < 0.05). The glucose tolerance of rats in the MBP group was significantly improved (*p* < 0.05). In diabetic mice, MBP significantly increased the activities of CAT, SOD, and GSH-Px. The inflammatory symptoms of liver cells and islet cells in the MBP group were alleviated, and the anti-inflammatory effect was partially correlated with the dose of MBP. After 4 weeks of treatment with MBP, the indices of blood lipid in the MBP group were significantly improved compared with those of the DM group (*p* < 0.05). Treatment with MBP (400 mg/kg) increases the levels of beneficial bacteria and decreases harmful bacteria in the intestinal tract of rats, thus altering the intestinal microbial community and antidiabetic effect on mice with T2DM by modulating gut microbiota. The findings suggest that MBP is a potential pharmaceutical supplement for preventing and treating diabetes.

## 1. Introduction

Millet (*Setaria italica*) is an important cereal crop cultivated in China. It provides nutrition and some health benefits for humans and animals [1]. As a traditional food in China, millet has high nutritional value. It is easy to digest and has various physiological functions such as gastrointestinal conditions [2,3,4]. Millet bran is a coarse food grain obtained from millets after husking; it is rich in bioactive substances such as cellulose, fibers, lipids, proteins, triglycerides, and minerals [5,6]. In recent years, several studies have reported that millet bran polyphenols exhibit high biological activities, including antioxidant [7], antitumor [8], immunomodulatory, antifungal, and hypoglycemic effects.

Polysaccharides are biological macromolecules found in almost all organisms. Natural polysaccharides exhibit important pharmacological activities without obvious side effects. They are non-cytotoxic, anti-inflammatory [9], antibacterial [10], anti-tumor [11,12], and hypolipidemic [13,14]. Polysaccharides have a preventive effect on certain diseases, such as diabetes and heart disease, which may increase the utilization of polysaccharides as food ingredients or pharmaceutical additives. Polysaccharides can directly or indirectly stimulate a variety of immune cells in mice, regulate the immune function of the body, and also exhibit hypoglycemic activity via intestinal flora in rats [15,16,17]. Increasing evidence suggests that polysaccharides may be an ideal functional food for the prevention or treatment of some diseases [18,19].

In recent decades, the number of patients with diabetes has been increasing, which is a serious threat to human health. Type 2 diabetes (T2DM), also known as non-insulin-dependent diabetes mellitus, is a chronic metabolic disease that causes hyperglycemia, insulin resistance, and low-grade inflammation, mainly affecting sugar, lipid, and protein metabolism [20,21]. T2DM is a gut microbiota-related disease and the first such association reported via a metagenomic study. Currently, the treatment of T2DM involves adjuvant therapy with insulin injection or hypoglycemic drugs to control blood sugar levels. Currently, the commonly used hypoglycemic drugs mainly include biguanides, acarbose, sulfonylureas, and thiazolidinediones. However, they are associated with limitations, such as exorbitant prices or serious side effects [22]. Intestinal flora exhibit metabolic and immune regulatory functions. Diabetes is a metabolic disease associated with the composition of intestinal flora. Regulating the composition of intestinal flora can alleviate the symptoms of diabetes and decrease the occurrence of complications in patients, which represents a new strategy to prevent and treat diabetes [9,17,23].

In this study, the antidiabetic effect of purified polysaccharide extracted from millet bran (MBP) was investigated in a T2DM model induced by streptozocin (STZ) combined with a high-fat diet. By analyzing the changes of indicators, weight, fasting blood sugar, and other bio-physiological parameters such as total cholesterol (TC), triglyceride (TG), superoxide dismutase (SOD), and catalase (CAT), combined with high-throughput sequencing, the effect of MBP on the intestinal flora in T2DM was elucidated. Thus, the positive regulatory function of polysaccharides on intestinal flora was determined. Further, a scientific basis for the use of millet bran in the treatment of T2DM was established.

## 2. Materials and Methods

### 2.1. Chemicals and Reagents

Millet bran (from Heyu yellow millet processing) was provided by Wufu Agricultural Products Development Co., Ltd. (Shanxi, China).

MBP was prepared via ultrasonic-enzyme-assisted extraction (UEAE). Briefly, the millet bran was defatted and powdered. The powder and multiple enzymes were soaked in deionized water and heated in a water bath, followed by ultrasonic treatment and centrifugation. The supernatant was concentrated, followed by extraction with four volumes of ethanol to facilitate precipitation at 4 °C overnight. The sediment was obtained, deproteinized, decolorized, and purified with DEAE-52 anion exchange and Sephadex G-100 column, followed by dialysis and lyophilization to obtain MBP. The MBP consisted of mannose, rhamnose, galactose, xylose, and arabinose in a molar ratio of 0.717: 0.592: 76.258: 1.036: 0.826, as reported in prior studies. Streptozotocin (STZ), hematoxylin, and eosin were purchased from Solarbio (Beijing, China). Chloral hydrate was purchased from Macklin (Shanghai, China). Paraformaldehyde was obtained from Servicebio Technology Co., Ltd. (Wuhan, China). High-fat diet (HFD) was purchased from Beijing Ke’ao Xieli Feed Co., LTD (Beijing, China). All commercial assay kits were purchased from Nanjing Jiancheng Bioengineering INST (Nanjing, China).

### 2.2. Animal Treatment

We purchased a total of specific pathogen-free 72 male Kunming rats, each weighing 22 ± 2 g (4 weeks) from SPF (Beijing, China) Biotechnology Co., Ltd. (Beijing, China). (Approval Code: SCXK (Jing) 2019-0010; Approval Date: 2021-03-19). This study was approved by the Safety Evaluation Center of the Chinese Academy of Radiation Protection in accordance with the National Institute of Health guidelines on the use of Laboratory Animals. All rats were caged (6 per cage) in an SPF animal laboratory at a constant temperature of 22 °C under a 12-h light/dark cycle. After 3 days of acclimatization, all rats except the normal control (NC) group (n = 12) were fed for 4 weeks with HFD, which consisted of a 67% basal diet with 2.5% cholesterol, 20% sucrose, 0.5% sodium cholate, and 10% lard. After feeding with HFD, all rats were fasted for 12 h and injected intraperitoneally with 40 mg/kg b.w. of 1% STZ-citrate buffer solution to increase blood glucose levels. The NC group was injected with citrate buffer. A fasting blood glucose (FBG) level greater than 11.1 mmol/L measured via the tail vein in the rats after 3 days and 7 days indicated the successful establishment of T2DM.

Diabetic rats were randomly divided into five groups (n = 12): T2DM (DM), positive control treated with metformin (MET, 200 mg/kg b.w.), diabetic rats treated with high-dose MBP (400 mg/kg b.w., MBP-400), diabetic rats treated with medium-dose MBP (200 mg/kg b.w., MBP-200) and diabetic rats exposed to low-dose MBP (100 mg/kg b.w., MBP-100). MBP-treated and MET groups received MBP and MET solutions, respectively. NC and DM groups were treated with an equal volume of normal saline for 4 weeks. Food intake was recorded weekly and body weights were measured daily to calculate the gavage dose. All rats were fasted for 12 h and narcotized with chloral hydrate (20%, intraperitoneal injection) at the end of the experiment. Blood samples were collected from the orbital and centrifuged (3000 rpm for 10 min, 4 °C) to acquire serum. Next, they were sacrificed by dislocation and dissected. The colon contents were placed in cryo-tubes, treated with liquid nitrogen, and stored at −80 °C immediately for gene sequencing and analysis of short-chain fatty acids (SCFAs). Liver, kidney, and pancreas tissues were derived from rats and immersed in paraformaldehyde for pathological section. One part of the liver was stored at −80 °C for further analysis.

### 2.3. Analysis of Parameters and Analytical Methods

#### 2.3.1. Fasting Blood Glucose (FBG) and Oral Glucose Tolerance Tests (OGTTs)

All rats were fasted for 8 h and the FBG levels were measured weekly with a Sinocare G-3 Blood Glucose Meter (Sinocare Inc., Changsha, China). The blood samples were derived from the tail vein. All rats were fasted at night. Free access to water was provided, and then we administered 20% glucose (2 g/kg, b.w.) by gavage. The blood samples were derived from the tail vein at 0, 30, 60, and 90 min.

#### 2.3.2. Biochemical Analysis

The biochemical indices including serum lipid parameters and anti-oxidative stress levels were determined. The levels of total cholesterol (TC), triglycerides (TG), high-density lipoproteins (HDL-C), and low-density lipoproteins (LDL-C) in blood serum were measured using a total cholesterol assay kit, A111-2-1; triglyceride assay kit, A110-2-1; high-density lipoprotein cholesterol assay kit, A112-2-1; and low-density lipoprotein cholesterol assay kit, A113-2-1 purchased from Nanjing Jiancheng Bioengineering INST, respectively.

The liver tissue was made into a 10% homogenate stock solution and diluted. The anti-oxidative stress indicators of the liver including superoxide dismutase (SOD), catalase (CAT), malondialdehyde (MDA), and glutathione peroxidase (GSH-Px) were measured using Superoxide Dismutase (SOD) assay kit, A001-3-2; Catalase (CAT) assay kit, A007-1-1; Malondialdehyde (MDA) assay kit, A003-1-2; and Glutathione Peroxidase (GSH-PX) assay kit, A005-1-2 purchased from Nanjing Jiancheng Bioengineering INST., respectively.

#### 2.3.3. Histopathological Analysis

After dissection, the liver, kidney and pancreas were quickly removed to obtain the tissue samples. The fresh tissue samples were soaked in paraformaldehyde solution for 48 h for fixation, followed by gradient elution with alcohol. They were then embedded in wax and sliced into 5-μm-thick sections, which were dried at 60 °C and stained. Sections were stained with hematoxylin, differentiated, and stained with anti-blue and eosin. The transparent specimens were then sealed and analyzed histopathologically under a light microscope [24,25].

#### 2.3.4. SCFA Analysis

The colonic contents were modified as described previously [26]. A mixture of standard solutions of acetic acid, propionic acid, n-butyric acid, i-butyric acid, n-valeric acid, i-valeric acid, and caproic acid dissolved in methanol to obtain 32 mmol/L, 16 mmol/L, 8 mmol/L, 4 mmol/L, 2 mmol/L, 1 mmol/L, 0.5 mmol/L, 0.25 mmol/L, and 0.1 mmol/L. The final concentration of the internal standard 2-ethylbutyric acid was 1 mmol/L. A 100-mg portion of rat feces was dissolved in 5 mL of methanol, vortexed, sonicated in an ice bath for 10 min, and centrifuged at 10,000 rpm for 10 min sequentially. The supernatant was retained after filtration through a 0.22 μm organic filter. 

The SCFAs in the samples were determined via Gas Chromatography (GC-2010ATF, 230 V, Shimadzu, Kyoto, Japan) under the following chromatographic conditions: DB-WAX quartz capillary column (30 m × 0.25 mm, 0.25 µm); initial temperature, 50 °C, held for 3 min, followed by heating at 6 °C/min to 120 °C, and heating at 10 °C/min to 220 °C for 5 min; carrier gas (N_2_) flow rate, 1.2 mL/min; separation ratio, 10:1; injection volume, 10 μL; inlet temperature, 230 °C; and detector temperature, 230 °C.

#### 2.3.5. 16S rRNA Gene Sequencing

Total genomic DNA was extracted from the feces using Magnetic Soil And Stool DNA Kit (Tiangen Biotech (Beijing) Co., Ltd., Beijing, China). The concentration and purity of DNA were tested on a 1% agarose gel. According to the concentration, DNA was diluted to 1 ng/µL with sterile water. The 16S rRNA genes in V4 regions were amplified with specific primers 515F(5′-GTGCCAGCMGCCGCGGTAA-3′) and 806R(5′-GGACTACHVGGGTWTCTAAT-3′). The barcode is a unique six-base sequence per sample [27]. All PCR mixtures contained 15 µL of Phusion^®^ High-Fidelity PCR Master Mix (New England Biolabs (Beijing) Ltd., Beijing, China), 0.2 µM of each primer, and 10 ng target DNA. The cycling conditions included a preliminary denaturation step at 98 °C for 1 min followed by 30 cycles at 98 °C (10 s), 50 °C (30 s), and 72 °C (30 s) and a final extension step at 72 °C for 5 min. An equal volume of 1x loading buffer (containing SYB green) was mixed with the PCR products and electrophoresed on a 2% agarose gel for DNA detection. The PCR products were mixed in equal proportions. The mixture of PCR products was purified using the Qiagen Gel Extraction Kit (Qiagen, Dusseldorf, Germany). The sequencing libraries were generated with NEBNext^®^ Ultra™ IIDNA Library Prep Kit (Cat No. E7645). The library quality was assessed on the Qubit@ 2.0 Fluorometer (Thermo Scientific, Waltham, MA, USA) and Agilent Bioanalyzer 2100 system according to the manufacturer’s recommendations. In the end, the library was sequenced on an Illumina NovaSeq platform, generating 250 bp paired-end reads [28].

#### 2.3.6. Bioinformatics Analysis

Original tags were merged using FLASH (Version 1.2.11, http://ccb.jhu.edu/software/FLASH/), followed by fastp (Version 0.20.0) analysis to obtain high-quality clean tags. The clean tags were then compared using Vsearch (Version 2.15.0) to detect the chimera sequences, which were then removed to obtain effective tags. To obtain the initial amplicon sequence variants (ASVs), the effective tags were denoised using the QIIME2 software (Version QIIME2-202006). The ASVs with an abundance of less than 5 were filtered out.

The representative sequences of each ASV were annotated. Meanwhile, ASVs are analyzed according to abundance and alpha diversity. A Venn diagram and petal diagram as well as the phylogenetic trees were constructed. The principal coordinates analysis (PCoA), principal component analysis (PCA), non-metric dimensionality scaling (NMDS), and other methods of dimensionality reduction and sample clustering were used to analyze the differences in community structure between samples or groups. In order to analyze the diversity, richness, and evenness of the communities among the grouped samples, methods such as Alpha Diversity and Beta Diversity including Observed_OTUs, Chao1, Shannon, Simpson, Dominance, Good’s coverage, Pielou, T-test, MetaStat, and LDA Effect Size (LEfSe) were used to test the significance of differences in species composition and community structure of the grouped samples. They were calculated based on weighted and unweighted unifrac distances in QIIME2 and visualized with R software (Version 3.5.). Finally, the PICRUSt2 (Version 2.1.2-b) software was used to predict the function of the microbial community in the ecological samples.

#### 2.3.7. Statistical Analysis

All graphs were exported using Origin 8.5. All data were presented as mean ± SD. All statistical analyses were managed using SPSS software. The *p*-values were 0.05 or 0.01 respectively, reflecting different levels of statistical significance.

The classify-sklearn module in QIIME2 (2021.2) software was used to obtain ASVs and then compared with the Silva138.1 database to determine the species of each ASV. Shannon, Simpson, Chao1, and other indices were calculated using QIIME2 software. The Rarefaction Curve and Species Accumulation Boxplot were drawn. QIIME2 software was used to calculate the Unifrac distance. The ade4 and ggplot2 packages from R (4.1.2) software were used to draw PCoA and NMDS dimensionality reduction pictures. LEfSe analysis was performed using the LEfSe software (http://huttenhower.sph.harvard.edu/lefse/), and the default LDA score threshold was set to 4. The function prediction was used by PICRUSt2 for analysis. Spearman correlation values of species and environmental factors were calculated and tested by corr.test function in the psych package of R, and the pheatmap function in the pheatmap package was used for visualization.

## 3. Results

### 3.1. The Effect of MBP on Body Weight and Food Intake

Taking T2DM rats as a research model, the regulatory effects of MBP on body weight and food intake were investigated. After continuous administration of different doses of MBP and MET, the changes in body weight and food intake of rats in each group are shown in Figure 1. Rats in the NG group were of normal body weight and slowly increased by 8.2%, compared with the NC group. The body weight of rats with STZ-induced T2DM fed with HFD decreased by 7.6% by the end of the administration. The body weight of rats treated with MBP and MET was increased. The MET and MBP-400 groups showed the maximum increase, reaching 6.5% and 4.1%, respectively, suggesting that MBP improved the loss in body weight of rats with T2DM. As shown in Figure 1B, the average daily food intake of DM was significantly higher than in the MET group. There was no significant difference in the average daily food intake of rats in the treated group, which indicated that MBP had no significant effect on the food intake of diabetic rats.

### 3.2. The Effect of MBP on FBG and Oral Glucose Tolerance

The regulating effect of MBP on blood glucose in rats with T2DM is shown in Figure 1C. Compared with the NC group, the FBG levels of rats were significantly higher in all the remaining groups of rats before administration (*p* < 0.05), which indicated the successful establishment of T2DM models. After MBP intervention, compared with the DM group, the FBG levels in MBP-400, MBP-200, and MBP-100 groups decreased by 40.40%, 32.64%, and 21.54%, respectively, and the level in the MET group was reduced by 41.28%. The results of OGTT are shown in Figure 1D. Oral intake of glucose solution led to a rapid increase in blood glucose concentration of the rats in each group. The blood glucose concentration of the rats in each group reached the maximum value at 30 min. Eventually, the blood glucose concentrations of the rats in each group declined. The blood glucose concentration of the rats in the DM group was higher than in the other groups at each point in time and continued to remain at a higher level. MBP intervention reduced the concentration of blood glucose below that of the DM group. The results show that MBP-400 and MBP-200 exhibit better hypoglycemic effects. MBP significantly improved glucose tolerance in rats with T2DM (*p* < 0.05). The results were similar to the results of tea polysaccharides. Treatment with high-dose TPS (*p* < 0.05) significantly reduced the FBG level, and the OGTT level of the rats declined after reaching the peak at 30 min [29]. Similarly, foxtail millet supplementation improved glucose metabolism in rats exposed to HFD and STZ-induced diabetes by lowing FBG and improving glucose tolerance [6].

### 3.3. The Effect of MBP on Serum Lipids in Rats with T2DM

Rats with T2DM manifest disorders of lipid metabolism. Therefore, in this study, the blood biochemical parameters (TC, TG, HDL-c, and LDL-c) of rats in different treatment groups were analyzed. As shown in Figure 2, compared with the NG group, the serum levels of TC, TG, and LDL-C in the DM group of rats were significantly increased (*p* < 0.05) by 100.4%, 221.7%, and 243.2%, respectively, HDL-C was significantly decreased (*p* < 0.05) by 55.45%, which indicated that the rats in the DM group developed symptoms of abnormal lipid metabolism. After four weeks of treatment with MBP solution, the blood lipids of diabetic rats were significantly improved. The levels of TC, TG, and LDL-C in the MBP intervention group were significantly decreased compared with the levels in the DM group. The serum levels of TC, TG, and LDL-C in the MBP-400 group were reduced by 38.62%, 45.94%, and 41.88%, and the levels of HDL-C increased (*p* < 0.05) by 78.42%. The results showed that MBP significantly improved symptoms of hyperlipidemia in rats with T2DM.

### 3.4. The Effect of MBP on Oxidative Stress in the Livers of Rats with T2DM

The endogenous antioxidant enzymes (CAT, SOD, GSH-Px, and MDA) constitute the first effective line of defense. By removing free radicals in the body, they prevent oxidative stress and cell damage and maintain the normal redox status of cells.

MDA levels are an important indicator of the degree of aging and oxidative damage in the body. As illustrated in Figure 3, compared with the NC group, the activities of CAT, SOD, and GSH-Px in the DM group of rats were significantly decreased (*p* < 0.05) by 35.40%, 65.20%, and 56.28%, respectively. MDA was significantly increased (*p* < 0.05) by 152.2%, which showed the successful establishment of the model. MBP-400 and MBP-200 were more effective in improving enzyme activity. The activities of CAT and SOD and GSH-Px in the MBP-400 group of rats were reduced (*p* < 0.05) by 26.32%, 82.43%, and 97.06%, respectively, while in the MBP-200 group, rats they were 19.54%, 77.12%, and 72.73%. Thus, there was no significant difference in the activity of SOD between the two groups.

### 3.5. Histopathological Analysis

The results of H&E staining of livers and pancreatic tissues of rats treated with polysaccharides in different treatment groups are shown in Figure 4. The liver structure of the NC group was regular. The cell structure was clear, with a neat arrangement, no steatosis, and less glycogen deposition. In the DM group, hepatocytes were severely swollen with blurred cellular edges, Glycogen deposits were seen in hepatocytes and the fat showed moderate lesions. Histopathology also revealed mild glycogen deposition in hepatocytes, occasional mild hypertrophy of hepatocytes, and very mild infiltration of punctate inflammatory cells. Intervention with different concentrations of MBP solution resulted in mild multifocal inflammatory cell infiltration, mild hypertrophy of hepatocytes, slightly clear cell boundaries, and mild glycogen deposition. Overall, a slight alleviation of symptoms was observed. Thus, MBP reversed the structural damage to liver tissues in diabetic rats.

As shown in Figure 5, the results of H&E staining of pancreatic tissue showed a clear cell structure and no pathological damage. The arrangement was very neat; the color of the nucleus was deep, and the number of cells was large in the NC group. However, in the DM group, the pancreatic tissue and the islet border were not clear. The islet cells were atrophied. The islet β-cells were disordered, and the pancreas was severely damaged. Further, the MBP-treated group showed a partial recovery of the islet tissue, a relative increase in the number of islet cells, and a clear and complete structure, which indicated that MBP improved the damaged tissue structure of islet cells in T2DM rats.

### 3.6. The Levels of SCFAs in Feces

We measured the concentration of SCFAs in feces obtained from colonic segments. The concentrations of acetic acid (AA), propionic acid (PA), isobutyric acid (iBA), butyric acid (BA), isovaleric acid (iVA), valeric acid (VA), caproic acid (CA), and total SCFAs are shown in Figure 6. The standard and samples are shown in Appendix A. The standard curve is presented in Appendix A. The levels of SCFAs are an important indicator of the intestinal microbial environment. The AA, PA, and BA concentrations constitute more than 96% of the SCFAs. The levels of each SCFA in the feces of rats in the DM group were significantly lower than in the NG group. After MBP treatment for 4 weeks, the total SCFA levels in the MBP-400, MBP-200, and MBP-100 groups increased. The levels of AA, PA, and BA were significantly increased compared with those in the other groups. The fecal AA level of T2DM rats in MBP-400 group was significantly higher than in MBP-200 and MBP-100 groups, with no significant differences in iBA, iVA, and CA. The differences in PA, iBA, and iVA levels between the MBP-400 and the MBP-200 groups were not statistically significant. Xia et al. reported that after gavage with Coix polysaccharide for 8 weeks, the SCFA levels were significantly increased in the group treated with the polysaccharides. The levels of AA, PA, BA, iBA, and iVA were significantly higher than in the DM group [30]. These data suggested significantly increased levels of fecal SCFAs in the MBP-treated rats.

### 3.7. Overall Structure of Gut Microbiota among Different Groups

We selected the V4 region of the 16S rRNA gene for sequencing. We generated a total of 3,661,969 raw reads with an average of 95,854 sequences per control sample, 101,242 sequences per MET sample, 103,917 sequences per high-dose sample; 102,389 sequences per medium-dose sample, and 103,090 sequences per low-dose sample. After chimera filtering, a total of 2,962,278 effective tags were obtained for 36 samples, with an average of 82,286 reads per sample. The average length of these sequences was 420–450 base pairs. In order to comprehensively evaluate the species richness and homogeneity within samples, the differences in microbial community structure between samples were analyzed using α-diversity and β-diversity indices [31]. In addition, the α-diversity metrics including Shannon, Simpson, Chao1 index, and rarefaction curve, and the β-diversity metric were evaluated using PCoA, PCA, and NMDS.

#### 3.7.1. Alpha Diversity

To elucidate the impact of MBP in terms of composition and abundance of intestinal flora in rats with T2DM, Venn analysis was performed on the community composition of the six groups (Figure 7A). The total number of ASVs was 420, while the number of ASVs unique to each group varied. The DM group had 193 unique ASVs; the MET group showed 217; the MBP-400 group had 690; the MBP-200 group showed 144; and MBP-100 carried 136 ASVs. The rarefaction curves are shown in Figure 7B. The tendency of curves demonstrated a high sampling coverage. Species richness was represented by the Observed index and Chao1 index; species diversity by the Shannon index and Simpson index while species evenness was represented by the Pielou index (Table 1). The Chao 1 indices of the MBP-400 and NC groups showed no significant difference. The observed species indices were significantly higher in the MBP-400 and MET groups than in the NC group (*p* < 0.01). Therefore, the MBP-400 group had a richer microbial community. Treatment with high-dose MBP and MET significantly increased the Shannon and Simpson indices compared with the NG group (*p* < 0.05), indicating that MBP-400 group had better community diversity and uniformity. Based on Pielou’ evenness index, treatment with MBP-400 and MET significantly improved the homogeneity of intestinal microbiota (*p* < 0.01).

#### 3.7.2. Beta Diversity

Beta diversity indicates a comparative analysis of the microbial community composition of different samples [32]. PCoA was based on the weighted unifrac distance and the unweighted unifrac distance, and the combination of principal coordinates with the greatest contribution was selected. Figure 7C shows the two main components of the fecal microbial community among the six groups, accounting for 25.19% and 14.23%, respectively, indicating a significant separation between the MBP-400, NC and DM groups. The clustered structure of the bacterial communities in each group was obvious, which indicated differences among the samples. Besides that, MBP-400 and NC groups were far from Axis 1 and close to Axis 2, which showed that MBP-400 and NC groups exhibited significant differences on PCoA 1. MBP-400 and NC groups were less pronounced on PCoA 2. The NMDS results are shown in Figure 7D. The ceftriaxone-treated rats exhibited distinct clustering with MBP-400 and NC groups. These findings indicate that 400 mg/kg of MBP solution altered the intestinal microbial community of rats.

### 3.8. Effects of MBP on the Species Composition of Intestinal Flora

To further elucidate the specific composition of microbial communities in different groups, we analyzed the differences at the phylum, class, order, family and genus levels. At the phylum level (Figure 8A), *Proteobacteria*, *Bacteroiaota*, *Firmicutes* and *Verrucomicrobiota* were the richest in the six groups, which also included *Desulfobacterota*, *Actinobacteria*, and *Campylobacter*. The abundance of *Proteobacteria* was significantly higher in MBP-400 than in the DM group, accounting for 34.93%. The members of *Bacteroiaota* showed insignificant changes with a slight decrease of 7.95%. This indicates that MBP enriched *Proteobacteria*, thereby affecting the metabolism of substances in rats. The abundance of *Firmicutes* and *Verrucomicrobiota* in the MBP-400 group accounted for 21.47% and 0.98%, which were considerably lower than 38.29% and 2.41%, respectively in the DM group. The NC group carried 0.19% of *Desulfobacterota* compared with 1.7% in the DM group. *Desulfobacterota* is a group of sulfate-reducing bacteria that convert sulfate into hydrogen sulfide, which can disrupt the integrity of the intestinal barrier. *Desulfobacterota* is also a Gram-negative bacterium, which is considered an opportunistic pathogen that mediates inflammatory diseases [33,34]. At the order level (Figure 8B), the gut microbiota in the six groups consisted of *Pseudomonadales*, *Bacteroidales*, *Oscillospirales*, *Peptostreptococcales-Tissierellales*, *Lachnospirales*, and *Verrucomicrobiales*. The abundance of *Pseudomonadales* was lower in the DM group than in NC and MBP-400 groups. *Pseudomonadales* play a useful role in promoting plant growth and controlling disease [35]. Further, the levels of *Peptostreptococcales-Tissierellales* and *Lachnospirales* were higher. At the genus level (Figure 8C), we also determined the average relative abundance of the highest 20 mesh: Compared with the DM group, the *Romboutsia* and *Bacteroides* were decreased in the MBP-400 group. The *Romboutsia* accounted for 6.17% in the DM group and 0.14% in the MBP-400 group. The *Romboutsia* genus is a rare anaerobic organism associated with some diseases [36]. The abundance of *Pseudomonas* and *Alloprevotella* in the DM group was remarkably lower than in the MBP-400 group. Wu et al. reported that the anti-diabetic mechanism of the traditional Chinese medicine compound Xiexin Decoction (XXT) was most likely to increase the abundance of some SCFA--producing bacteria and anti-inflammatory bacteria, such as *Alloprevotella*, which synthesize caprylic acid [37]. The relative abundance of *Akkermansia* and *Alistipes* in the MBP-treated group was higher than in that in the DM group and accounted for 6.41% and 2.75% in the MBP-200 group, and 0.95% and 0.75% in the MBP-400 group. The probiotic role of *Akkermansia* in metabolic regulation, immune regulation, and gut health protection has been extensively studied [38]. *Alistipes* are associated with the release of gut hormones that regulate insulin release and reverse insulin resistance [39]. Therefore, we speculate that the MBP-200 group was better at regulating gut microbiota. The *Lactobacillusn* and *Lachnospiraceae*_NK4A1 36_group belong to the *Lachnospira* family, phylum *Firmicutes phylum*, and their abundance in the MBP-treated group was a little higher than in the DM group. Relevant studies have shown that members of the *Lachnospira* family in the human gut commensal flora express two “superantigens”, which activate the IgA response in the body and play an important role in intestinal homeostasis [40]. In conclusion, the MBP supplementation altered the species composition of intestinal flora in rats.

We ranked the top 30 flora at the genus level and drew a heat map by clustering the abundance similarity of different groups (Figure 8D). The differences in color reflect the similarities and differences in the composition of the intestinal flora of rats in each group. Significant differences existed between the groups. *Lactobacillus*, *Acinetobacter*, *Faecalibacterium*, and *Alcanivorax* were the dominant strain in the MBP-400 and *Alistipes* was predominantly enriched in the MBP-200 group. *Alistipes* are mainly present in the gut of healthy individuals from an ecological perspective, and it produces SCFAs26. The intestinal microbiota of different groups of rats are similar but also show some differences. The intestinal flora of rats fed with HFD gradually deviated from normal, and MBP had a positive effect on the regulation of intestinal flora in rats.

### 3.9. Identification of Phenotypic Biomarkers

In order to determine the statistically significant differences between groups, we used LEfSe, a tool used for the discovery and interpretation of high-dimensional biomarkers. The LDA value distribution histogram shows species with an LDA score greater than 3, that is, biomarkers with statistically significant differences between groups (Figure 8E). In a cladogram, the circles symbolize the taxonomic level from phylum to genus (or species) (Figure 8F). According to the results shown in Figure 8E, the phenotypic biomarkers in NC, DM, MET, MBP-400, MBP-200, and MBP-100 were identified for landmarks 3, 6, 8, 3, 3, and 0. The largest genera were enriched in the MBP-200 and MET groups. Species that have a greater impact on community structure in the NC group were o_*Clostridia*_UCG_014, g_*Clostridia*_UCG_014, and f_*Clostridia*_UCG_014 while those in the DM group were o_*Peptostreptococcales_Tissierellales*, f_*Peptostreptococcaceae*, g_*Romboutsia*, and c_*Bacilli*. Those in the MET group were p_*Firmicutes*, c_*Clostridia*, o_*Lachnospirales*, and f_*Lachnospiraceae*. Species with a greater impact on community structure in the MBP-400 group were o_*Lactobacillales*, f_*Lactobacillaceae*, and g_*Lactobacillus* while those in the MBP-200 group were g_*Pseudomonas*, f_*Pseudomonadaceae*, and o_*Pseudomonadales*. The MBP-100 group is not presented in the figure because the LDA value is less than 4, and no statistically significant biomarker was detected for this group. As shown in Figure 8F, the signature microorganisms of the MBP-200 group and other groups were derived from different phylum. The o_*Clostridia*_UCG_014 and f_*Clostridia*_UCG_014 showed the highest abundance in the NC group; o_*Lachnospirales*, f_*Lachnospiraceae*, f_*Eubacterium*_coprostanoligenes_group and f_*Oscillospiraceae* in the MET group; and o_*Peptostreptococcales_Tissierellales* and f_*Peptostreptococcaceae* in the DM group. The highest abundance of microorganisms in the MBP-400 group involved o_*Lactobacillales* and f_*Lactobacillaceae*. All of the above microorganisms belong to the p_*Firmicutes*. *Lactobacillales* belong to *Enterococcus*, which is a probiotic that enhances the activity of macrophages, promotes the immune response of animals, and increases antibody levels [41]. The above results indicate that MBP increases the levels of beneficial bacteria in the intestinal tract of rats, and potentially alters the intestinal microbial community.

### 3.10. Correlation Analysis of Intestinal Flora and Physiological Indicators

Based on these significantly altered gut bacteria in T2DM, we analyzed their relationship with markers of diabetes via Spearman correlation. As shown in Figure 8G, *Faecalibaculum*, *Desulfovionio*, and *Dubosiella* are significantly positively correlated with TC and TG, and *Helicobacter* and *Alloprevoiella* are significantly correlated negatively. *Gastranaerophilales* and *Dubosietla* are significantly positively correlated with LDL-C, unlike *Clostridia*_UCG.014, which is negatively correlated. Further, *Clostridia*_UCG:014 is a probiotic with a significant positive correlation with CAT, GSH-Px, and SOD, but a negative correlation with MDA. *Clostridia*_UCG:014 is effective in protecting against liver injury in rats [42]. These results suggest that abnormal changes in gut microbiota may exacerbate the development of diabetes.

Similarly, the effects of *Lycium barbarum* Polysaccharide (LBP) on T2DM in mice were mediated via modulation of gut microbiota. The results showed that LBP notably improved the composition of intestinal flora, increasing the relative abundance of *Bacteroides*, *Ruminococcaceae*_UCG-014, *Intestinimonas*, *Mucispirillum*, and *Ruminococcaceae*_UCG-009, and decreasing the relative abundance of *Allobaculum*, *Dubosiella*, and *Romboutsia* [16].

### 3.11. Functional Prediction

PICRUSt2 is a bioinformatic software package used to predict the metagenomic function based on 16S rRNA. Sequences of ASVs obtained from 16S rRNA sequencing of each sample were localized to the KEGG pathway to predict changes in metabolic function within the microbiome under PEP regulation. Figure 8H presents a statistical analysis of the results of gene prediction. Metabolism, Environmental Information Processing, and Genetic Information Processing are the major metabolic pathways of intestinal flora. Carbohydrate metabolism, amino acid metabolism, and energy metabolism are the main metabolic functions.

## 4. Conclusions

The results of this study showed that treatment with MBP reduced blood sugar levels, improved serum lipid metabolism, and enhanced hepatic antioxidant capacity. It suggested that MBP alleviated symptoms of HFD-induced T2DM. During gavage, the FBG of the rats in the MBP group was significantly reduced (*p* < 0.05). We observed the trend of OGTT in each group and found that the tolerance of the rats in the MBP group to glucose was significantly improved (*p* < 0.05). The histopathological analysis of the liver and pancreas reveals that the inflammatory symptoms of the liver and islet cells in the MBP group were alleviated. The cellular hypertrophy was also alleviated, indicating the therapeutic effect of MBP. After 4 weeks of treatment with MBP, serum TG, TC, and LDL-C levels were significantly decreased, whereas HDL-C levels were significantly increased. The levels of CAT SOD GSH-PX in the liver were significantly increased. The MDA levels were also significantly increased (*p* < 0.05). The SCFA levels of the MBP group measured via gas chromatography were significantly higher than in the DM group (*p* < 0.05). A higher dosage of MBP enhanced the hypoglycemic effects.

In addition, the analysis of 16S rDNA revealed that MBP notably improved the composition of intestinal flora, increasing the relative abundance of beneficial bacteria and decreasing harmful bacteria in the intestinal tract of rats. At the genus level, the relative abundance of *Akkermansia* and *Alistipes* in the MBP-200 group was higher than in the MBP-400 group, suggesting that the MBP-200 group regulated gut microbiota more effectively. The above results indicated that MBP plays an active role in regulating the intestinal flora of rats with T2DM. The results of this study provide a theoretical foundation and a scientific basis for the comprehensive application of millet bran in the treatment of T2DM.

## Figures and Tables

**Figure 1 foods-11-03406-f001:**
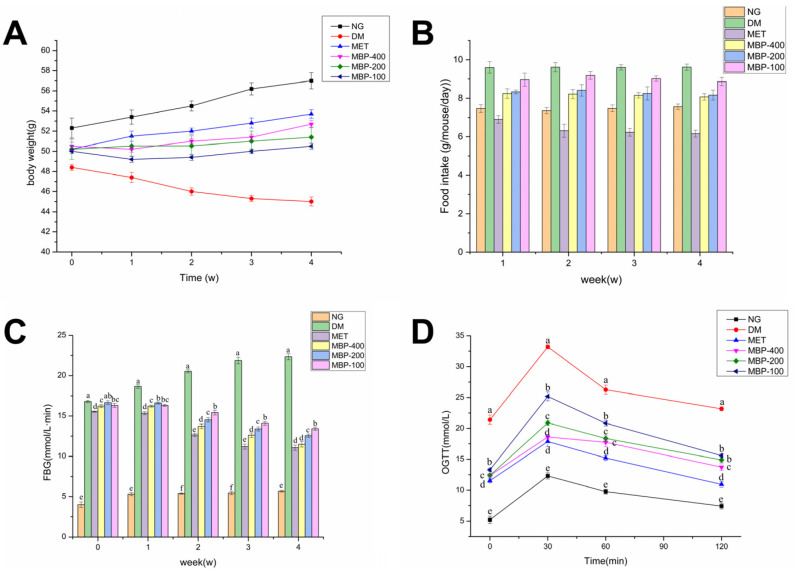
Effects of MBP on body weight (**A**), food intake (**B**), fasting blood glucose (FBG) (**C**), and oral glucose tolerance test (OGTT) (**D**). Different letters are significantly different at *p* < 0.05.

**Figure 2 foods-11-03406-f002:**
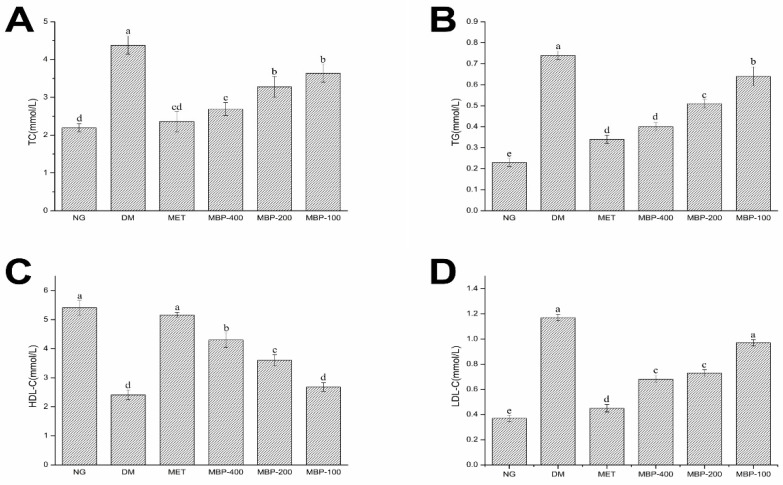
Serum levels of total cholesterol (TC) (**A**), triglycerides (TG) (**B**), high-density lipoproteins (HDL-C ) (**C**) and low-density lipoproteins (LDL-C) (**D**) in rats. Different letters are significantly different at *p* < 0.05.

**Figure 3 foods-11-03406-f003:**
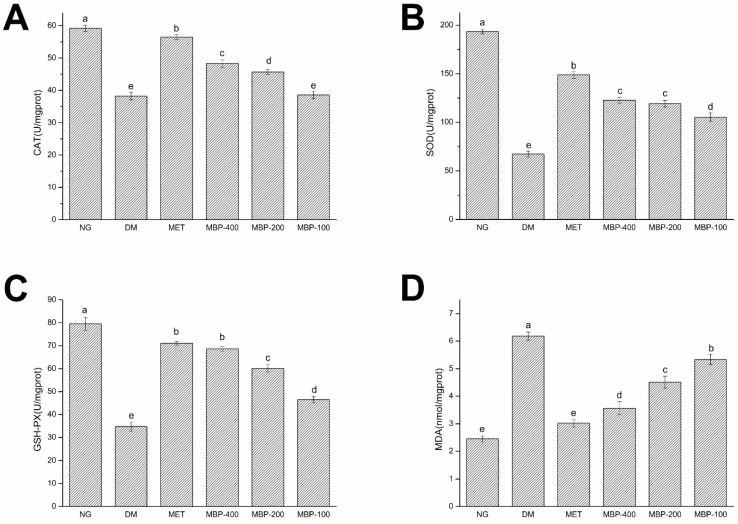
Expression of catalase(CAT) (**A**), superoxide dismutase (SOD) (**B**), glutathione peroxidase( GSH-Px) (**C**), and malondialdehyde (MDA) (**D**) in liver of rats. Different letters are significantly different at *p* < 0.05.

**Figure 4 foods-11-03406-f004:**
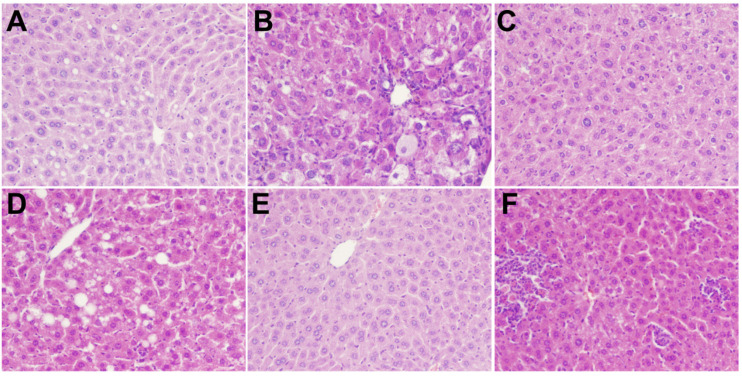
Effects of MBP on histopathological changes in liver: NC group (**A**), DM group (**B**), MET group (**C**), MBP-100 group (**D**), MBP-200 group (**E**), MBP-400 group (**F**).

**Figure 5 foods-11-03406-f005:**
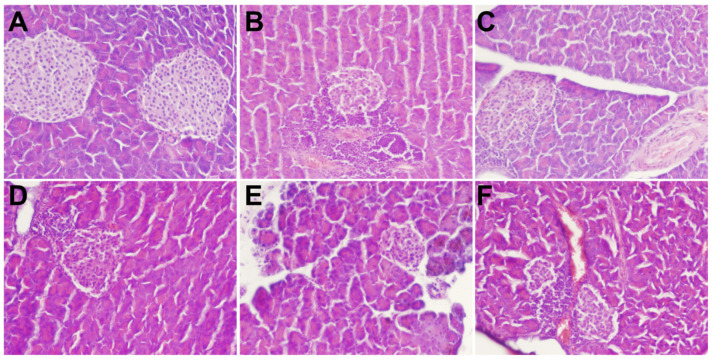
Effects of MBP on histopathological changes in pancreas: NC group (**A**), DM group (**B**), MET group (**C**), MBP-100 group (**D**), MBP-200 group (**E**), MBP-400 group (**F**).

**Figure 6 foods-11-03406-f006:**
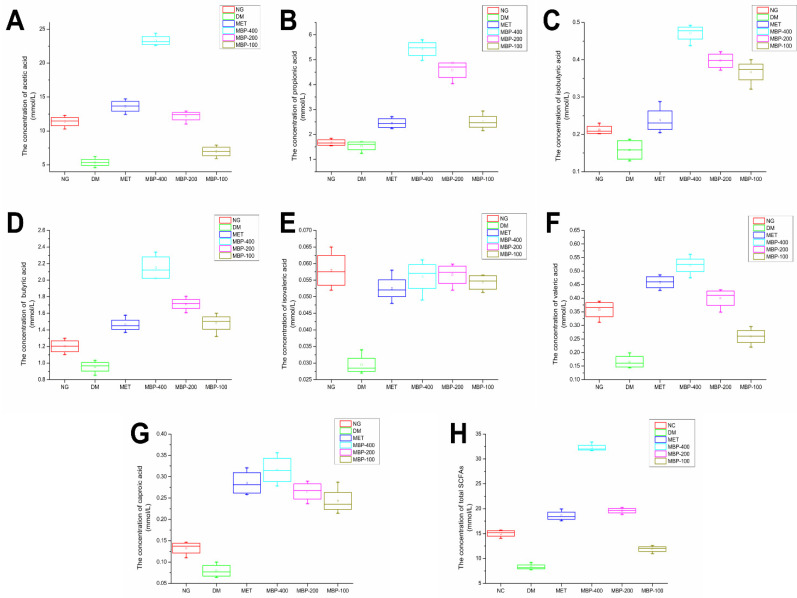
Levels of SCFAs in six groups, AA (**A**), PA (**B**), iBA (**C**), BA (**D**), iVA (**E**), VA (**F**), CA (**G**) and total SCFAs (**H**).

**Figure 7 foods-11-03406-f007:**
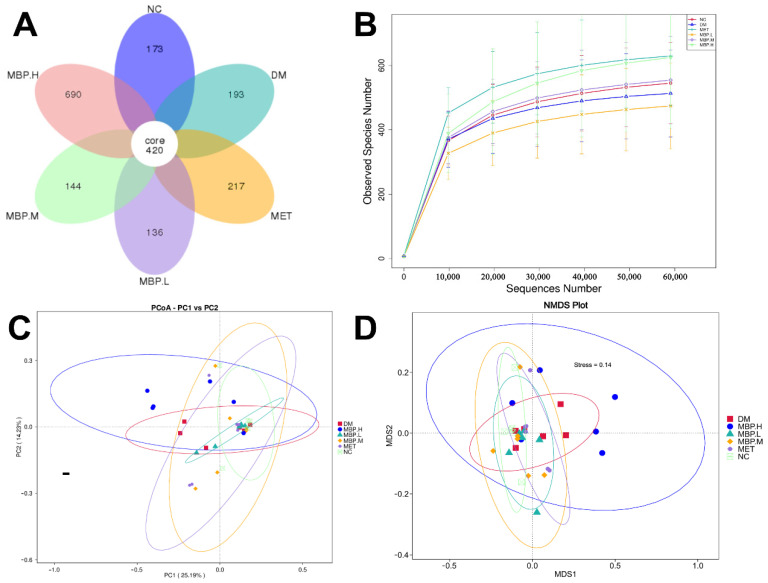
Flower figure (**A**), Rarefaction analysis (**B**), Unweighted UniFrac-based PCoA (**C**) and NMDS analysis (**D**).

**Figure 8 foods-11-03406-f008:**
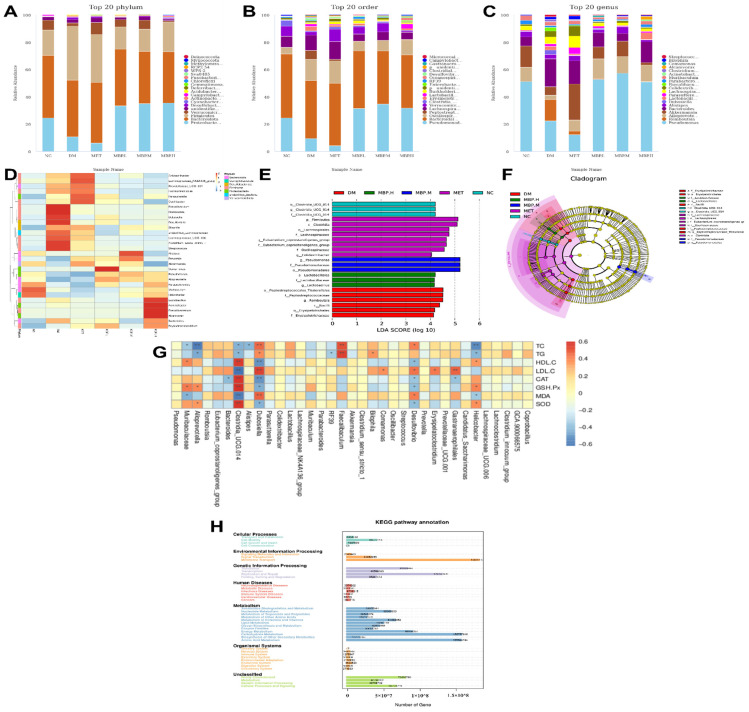
Relative abundance of microbiota at the levels of phylum (**A**), order (**B**), and genus (**C**). Heatmap analysis of fecal microbiota at genus level (**D**). Histogram of LDA effects of biomarker species (**E**). The taxonomic diagrams obtained from LEfSe analysis (**F**). Spearman correlation (**G**) and KEGG analysis of relative abundance of metabolic pathways (**H**).

**Table 1 foods-11-03406-t001:** MBP regulates alpha diversity of gut microbiota in rats.

Index	NC	DM	MET	MBP-400
Observed species	487.16 ± 107.84 *	456.83 ± 73.02	586.17 ± 99.58 **	620.50 ± 137.25 **
Chao 1	499.28 ± 102.04	473.97 ± 96.98	588.27 ± 100.61	506.44 ± 175.98
Shannon	6.21 ± 0.57	5.26 ± 1.10	6.51 ± 0.61 *	6.32 ± 0.71 *
Simpson	0.90 ± 0.09	0.76 ± 0.12	0.91 ± 0.03 *	0.98 ± 0.03 *
Pielou’ evenness	0.59 ± 0.22 *	0.53 ± 0.20	0.71 ± 0.05 **	0.76 ± 0.07 **

* Indicates a significant difference (*p* < 0.05); ** Indicates that the difference is extremely significant (*p* < 0.01).

## Data Availability

Data is contained within the article.

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
