# Peer review of "Antidiabetic Effect of Millet Bran Polysaccharides Partially Mediated via Changes in Gut Microbiome"

_foods, 2022, doi:10.3390/foods11213406_

Round 1
Reviewer 1 Report
Please add aim and importance of your research in starting line of abstract.
Authors have mention six groups while treatment has been mentioned for three groups 400, 200, 100 mg/kg. Please elaborate.
Please add some quantitative results in the abstract for better understanding as abstract is your main parameter that appear on the top of your work.
Authors should add extraction methods of polysaccharides or characterization method of microbiota in short way.
Add conclusion in the end of abstract.
Authors should must add proper reasoning and importance of this article in the form of rationale at the end of introduction.
Author should mentioned about the varieties of millet as a lot of Millet varieties has been produced and there are chances to have different effects of different varieties.
Please remove these lines these are for your understanding not for part of manuscript.
.2. Materials and Methods. 62 The Materials and Methods should be described with sufficient details to allow oth- 63 ers to replicate and build on the published results. Please note that the publication of your 64 manuscript implicates that you must make all materials, data, computer code, and proto- 65 cols associated with the publication available to readers. Please disclose at the submission 66 stage any restrictions on the availability of materials or information. New methods and 67 protocols should be described in detail while well-established methods can be briefly de- 68 scribed and appropriately cited. 69 Research manuscripts reporting large datasets that are deposited in a publicly avail- 70 able database should specify where the data have been deposited and provide the relevant 71 accession numbers. If the accession numbers have not yet been obtained at the time of 72 submission, please state that they will be provided during review. They must be provided 73 prior to publication. 74 Interventionary studies involving animals or humans, and other studies that require 75 ethical approval, must list the authority that provided approval and the corresponding 76 ethical approval code.
Methodology is not clear, most of the methods have no sequence please set according to the research plan.
Discussion needs serious attentions, there are no proper reasoning and a comparison in any of the parameter.
Please add some latest literature to improve the quality of article.
Conclusion is too much lengthy please reduce it and add only important parameters in it.
Please thoroughly check all the references according to the journal format.
Reviewer 2 Report
The study reported by the authors was impeccably designed and executed and the results and discussion are unbiased and evidence-based. Certain modifications are suggested to improve even more the scientific soundness, attractiveness, and uniqueness of the manuscript:
General
· The manuscript will improve even more if the English grammar (e.g. rarefaction?- Figure 7) and style are reviewed once again by a native English-spoken person or by a formal translation agency.
· The meaning of all abbreviations should be clarified the first time they are mentioned.
Title. OK, although it could be more insightful: "The antidiabetic effect of millet bran polysaccharides is partially due to gut microbiome changes"
Abstract. This section should be more quantitative> qualitative, highlighting specific dose-dependent effects of Millet bran polysaccharides.
Introduction. A) This section should highlight the scientific contribution of the study, particularly as to the functional/nutraceutical value of millet (see doi: 10.1016/j.procbio.2021.08.011, 10.9755/ejfa.v25i7.12045 , 10.1080/10942912.2019.1668406) and its GI fermentable products (postbiotics: ) 10.1111/jfbc.12859. B) Moreover, very recently Ren et al. (2022; doi: 10.1016/j.fshw.2021.07.013) published a quite similar experimental design whose results point in the same direction, you should discuss the contribution of your study vs. this reference and any other recently published. C) Delete Food´s instructions (“2. Materials and methods” Line 62, whole paragraph between lines 63-77).
Methods. A) Section 2.2: Provide details on ethics (approval number, date, etc.), provide dose and route of administration of chloral hydrate, provide a scheme of STZ+HFD induction protocol and composition of diets (see as an example: doi: 10.1016/j.fshw.2021.07.013) B) Section 2.3: Certain portable glucose meters tend to be inaccurate, provide details as to its sensitivity, and calculation of glycemic AUC is strongly needed (result/discussion as well). C) Section 2.4: Change “Biomedical analysis” for “Biochemistry” “with a kit” should be described in detail when needed (kit name, brand, part number, etc). D) Section 2.5: Provide more details (e.g., magnification, inspection strategy, etc. See Doi: 10.1016/j.fshw.2021.07.013). E) Section 2.6: provide details of the GC-MS equipment. F) Section 2.9: Expand details and consider performing linear/logistic regression analysis, correlation matrix, etc. as complementary inductive methods.
Results & Discussion. A) A very succinct (not in-depth) discussion was described in certain sections [e.g. Section 3.2. just three lines (232-234), section 3.10 no discussion at all]. B) Each subsection should be discussed in a more comparative & inductive (explain phenomena, hypothesis) way. C) Section 3.9 is very hard to read/follow.
Tables & Figures. A) Please include statistically significant differences between groups. B) Tables should be formatted as customary (see Food´s guidelines/ other recent articles). C) All figures should be provided with a higher resolution (>300 dpi). D) Figure 1: It is suggested to include (or replace) and discuss a Food Efficiency Ratio [FER= weight gain (g)/ food intake (g)]. E) Figures 2, 3, and 6 (end-point values) could be included in just one table. F) When needed (e.g. Figure 8) split figures and supply any piece (figure) as supplementary material.
Conclusion. OK
References. Check once again for any unformatted or not properly cited references.
Reviewer 3 Report
The manuscript entitled “Effects of polysaccharide from millet bran on Type 2 diabetic rats via Gut Microbiota and Metabolism Alteration” studied the regulatory effect of millet bran on intestinal microbiota in model of type 2 diabetes using rats as a model. Authors showed that the occurrence of T2DM could lead to the imbalance of intestinal microbiota in examined rats whereas application of millet bran in the intervention treatment of type 2 diabetes could be useful.
Page 2, line 49 – please rewrite to make more sense “Diabetes as a kind of metabolic disease that interacts with intestinal flora”
Page 2, line 53 – please rewrite the whole paragraph explaining the aim of the study since it is not clear in its present form. This especially goes for the first sentence of the paragraph.
Page 2, line 62 – please erase 2. Materials and methods
Page 2, line 63 to 77 – you have left the text from the template and this should be erased.
Page 3, line 132 – this sentence is not clear. At the end should be “were measured” if these parameters were measured in rat’s liver?
Page 7, line 254 – please indicate which enzymes were measured and do not use etc., be specific (CAT, SOD and GSH-Px, etc.)
In the section 3.4. The effect of MBP on the oxidative stress in the liver of T2DM rats please cite the statements regarding different enzymes and MDA that you are describing.
Page 17, line 483 – “induced by T2DM”?
Minor remarks:
Please provide full word for T2DM in Abstract section and on the first mention in the second paragraph of the Introduction section.
Latin names of species should be written in italics.
Page 3, line 109 – please explain abbreviation MET on the first mention. I guess this is metformin?
Please be consistent with using h and/or hours etc.
Please explain abbreviation in the Figures legends. Please use larger font in the Figures (especially in the X and Y axes). Figure 6 is almost unreadable as well as Figure 8.
Figure 7 – please write F in flower with the capital letter.
Although not my mother tongue in my opinion the paper would benefit from English language editing since it is a bit hard to follow in its present form.
Round 2
Reviewer 1 Report
I think English grammar is still poor authors should consult with a native english speaker.
Overall i am satisfied,but it needs to improve the language.
Reviewer 3 Report
Authors responded to the raised questions/suggestions and changed their paper accordingly.
